# Acute Kidney Injury Following Admission with Acute Coronary Syndrome: The Role of Diabetes Mellitus

**DOI:** 10.3390/jcm10214931

**Published:** 2021-10-25

**Authors:** Arthur Shiyovich, Keren Skalsky, Tali Steinmetz, Tal Ovdat, Alon Eisen, Abed Samara, Roy Beigel, Sagi Gleitman, Ran Kornowski, Katia Orvin

**Affiliations:** 1Rabin Medical Center, Department of Cardiology, Faculty of Medicine, Tel-Aviv University, Tel Aviv 69978, Israel; kskalsky@gmail.com (K.S.); alon201273@gmail.com (A.E.); AbedS2@clalit.org.il (A.S.); ran.kornowski@gmail.com (R.K.); katiaorvin@gmail.com (K.O.); 2Rabin Medical Center, Department of Nephrology, Faculty of Medicine, Tel-Aviv University, Tel Aviv 69978, Israel; talisht@clalit.org.il; 3Sheba Medical Center, Department of Cardiology, Faculty of Medicine, Tel-Aviv University, Tel Aviv 69978, Israel; Tal.Cohen@sheba.health.gov.il (T.O.); Roy.Beigel@sheba.health.gov.il (R.B.); 4Baruch Padeh Medical Center, Division of Cardiovascular Medicine, Poriya, Tiberias 1528001, Israel; gl.sagi@gmail.com

**Keywords:** acute kidney injury, diabetes mellitus, acute coronary syndrome

## Abstract

Purpose: To evaluate the role of diabetes mellitus in the incidence, risk factors, and outcomes of AKI (acute kidney injury) in patients admitted with ACS (acute coronary syndrome). Methods: We performed a comparative evaluation of ACS patients with vs. without DM who developed AKI enrolled in the biennial ACS Israeli Surveys (ACSIS) between 2000 and 2018. AKI was defined as an absolute increase in serum creatinine (≥0.5 mg/dL) or above 1.5 mg/dL or new renal replacement therapy upon admission with ACS. Outcomes included 30-day major adverse cardiovascular events (MACE) and 1-year all-cause mortality. Results: The current study included a total of 16,879 patients, median age 64 (IQR 54–74), 77% males, 36% with DM. The incidence of AKI was significantly higher among patients with vs. without DM (8.4% vs. 4.7%, *p* < 0.001). The rates of 30-day MACE (40.8% vs. 13.4%, *p* < 0.001) and 1-year mortality (43.7% vs. 10%, *p* < 0.001) were significantly greater among diabetic patients who developed vs. those who did not develop AKI respectively, yet very similar among patients that developed AKI with vs. without DM (30-day MACE 40.8% vs. 40.3%, *p* = 0.9 1-year mortality 43.7 vs. 44.8%, *p* = 0.8, respectively). Multivariate analyses adjusted to potential confounders, showed similar independent predictors of AKI among patients with and without DM, comprising; older age, chronic kidney disease, congestive heart failure, and peripheral arterial disease. Conclusions: Although patients with DM are at much greater risk for AKI when admitted with ACS, the independent predictors of AKI and the worse patient outcomes when AKI occurs, are similar irrespective to DM status.

## 1. Introduction

The prevalence of diabetes mellitus (DM), a major cardiovascular risk factor, is increasing worldwide [1,2]. Among patients presenting with an acute coronary syndrome (ACS), those with DM are at increased risk of in-hospital morbidity and mortality compared with those without DM [3,4]. Acute kidney injury (AKI) is a common complication in patients presenting with acute coronary syndrome (ACS), particularly following percutaneous coronary intervention (PCI) [5,6,7,8,9]. Furthermore, when AKI occurs in patients with ACS, it is associated with significantly worse short- and long-term outcomes that include increased risk for renal replacement therapy, prolonged hospitalization, greater mortality and economic burden [5,6,7,8,9,10]. Patients with DM are considered to be at increased risk for AKI in the setting of ACS and PCI [10,11,12,13]. Moreover, the treatment for AKI in this scenario is rather preventive and consists mostly of supportive care or hemodialysis. However, data evaluating and directly comparing (from a single cohort) the incidence, outcomes, and the prognostic markers of AKI among patients, with and without DM, are sparse.

The aim of the current study was to evaluate the role of diabetes mellitus in the incidence, risk factors, and outcomes of AKI in patients admitted with ACS.

## 2. Methods

### 2.1. Study Design and Population

The current study included 16,879 consecutive patients from the ACS Israeli Surveys (ACSIS) between 2000 and 2018. Details of the registry have been previously reported [14,15]. In shortly, ACSIS is a biennial prospective national registry of all patients with ACS hospitalized in 26 coronary care units and cardiology departments in all general hospitals in Israel over a 2-month period (March to April). Data were recorded on prespecified forms for all admitted patients diagnosed with ACS. Admission and discharge diagnoses were recorded by the attending physicians. Patient management was at the discretion of the attending physicians. All patients signed an informed consent form for participating in the ACSIS registry at each medical center, and each institution received the approval of its institutional review board (ethics committee). AKI was defined as an absolute increase in serum creatinine (≥0.5 mg/dL) or above 1.5 mg/dL (when no prior level existed in patients without a diagnosis of chronic kidney disease) or new renal replacement therapy upon admission with ACS. It should be stated that a gold standard for the definition of AKI and the ideal margins of absolute and relative increases in serum creatinine used for the definition is controversial [16]. As our study included not only contrast-induced AKI (CI-AKI) but rather all AKI in the context of ACS, even fewer data exists about the optimal definition. An increase of 0.5 mg/dL is one of the previously reported and accepted definitions that have been shown to be quite accurate [16,17] and is not the most strict or lenient, hence, it was chosen by us for this study. Regarding patients without a recent previous creatinine value, who did not have a diagnosis of chronic kidney disease, we chose the definition of creatinine above 1.5 mg/dL based on the previously reported definition of the relative increase in serum creatinine to ≥1.5 times baseline [18] assuming the baseline to be 1 mg/dL (which is close to the upper limit of normal values in both sexes). We believe using this definition would be more accurate than excluding these patients (due to missing values) and causing a potentially significant selection bias.

Following comparison between patients with vs. without AKI, the study cohort was divided into four groups according to the DM and AKI to enable status as following:
no-AKI_no-DM: patients without DM that did not develop AKI.no-AKI_DM: patients with DM that did not develop AKI.AKI_no-DM: patients without DM that developed AKI.AKI_DM: patients with DM that developed AKI.

These subgroups enabled the performance of analyses intended to better and more accurately separate and investigate the role of DM among patients with ACS who developed AKI.

### 2.2. Outcomes

The outcomes included 30-day major adverse cardiovascular events (MACE), comprised of: all-cause mortality, myocardial infarction (MI), stroke, unstable angina, stent thrombosis, and urgent revascularization. An additional outcome was 1-year all-cause mortality. Data regarding the outcomes were determined by hospital chart review, telephone contact, clinical follow-up, and by matching identification numbers of patients with the Israeli National Population Registry (for 30-day and 1-year mortality).

### 2.3. Statistical Analysis

Patients’ characteristics are presented as median (IQR) for continuous variables and as frequency (%) for categorical variables. Comparisons between the study groups were tested with chi-square for categorical variables and with Mann–Whitney–Wilcoxon test for non-normally distributed continuous variables. The normality of continuous variables was assessed using Shapiro–Wilk test. Survival curves were plotted, and the Kaplan–Meier log-rank test was used to test the variable of interest on survival.

Logistic regression models were used to assess the relationship between patients’ baseline characteristics and the outcome of developing AKI, and Cox proportional hazards regression models were used for the outcome of 1-year mortality among the whole cohort and among the subsets of diabetic and non-diabetic patients.

All tests were conducted at a two-sided overall 5% significance level (*p* = 0.05). Statistical analyses were performed using R software.

## 3. Results

The study cohort included a total of 16,879 patients, median age 64 (IQR 54–74), 77% males, 36% with DM. The overall incidence of AKI was 6% (1016 patients), significantly higher among patients with DM vs. without DM (8.4% vs. 4.7%, *p* < 0.001). Comparison of baseline characteristics according to AKI in the entire cohort are presented in the Appendix A. A multivariate analysis (Appendix A) showed that DM was independently associated with AKI (adjusted odds ratio (OR) 1.4, 95% confidence interval (CI) 1.2–1.6, *p* < 0.001). The baseline characteristics of the patients with and without DM by AKI are presented in Table 1. Compared to patients without AKI, those who developed AKI (diabetics and non-diabetic) were older, mostly females, with a higher rate of chronic kidney disease, hypertension, and congestive heart failure and presentation as STEMI, yet lower rate of smokers and positive family history. Furthermore, patients who developed AKI had greater rates of prior myocardial infarction and prior coronary artery bypass graft surgery, and higher rate of treatment with ACE inhibitors/ARBs, nitrates, and diuretics compared with patients that did not develop AKI.

As presented in Table 2, The clinical presentation and most in-hospital complications (e.g., pulmonary edema, cardiogenic shock, mechanical and arrhythmic complications) were more severe/prevalent among patients with AKI (both diabetics and non-diabetics) vs. patients who did not develop AKI. However, the rates of coronary angiography and PCI were lower among patients with AKI (among patients with and without DM).

Table 3 and Figure 1 present the 30-day MACE and 1-year mortality according to DM status and the occurrence of AKI. The rates of 30-day MACE and nearly all its components, as well as the rates of 1-year mortality, were significantly greater among patients who developed AKI. However, the rates of the latter outcome were similar among patients with AKI irrespective of their DM status (diabetics = 43.7%, non-diabetics 44.8%). Additionally, multivariable analysis adjusted to potential confounders (Appendix A) shows no statistically significant interaction between DM and AKI in the prediction of 1-year mortality (*p* = 0.1).

Additionally, a comparison of the baseline characteristics between patients with and without DM is presented in Appendix A.

Multivariable analyses, including the patients’ baseline characteristics (Figure 2), showed similar independent predictors of AKI among patients with and without DM comprising older age, chronic kidney disease, peripheral vascular (arterial) disease, and congestive heart failure.

Furthermore, a multivariable analysis with 1-year mortality as the outcome (Appendix A) showed similar predictors among patients with and without DM, specifically AKI was the strongest independent predictor in both groups (with DM: OR = 3.55, 95% CI: 2.99–4.21, *p* < 0.001, without DM: HR = 3.69, 95% CI: 3.09–4.4, *p* < 0.001) with similar HR in both.

## 4. Discussion

In the current study, we have shown that although patients with DM have a greater risk for developing AKI when admitted with ACS, other independent predictors of AKI and the worse outcomes associated with AKI are similar among diabetics and non-diabetic patients.

The observed prevalence of DM in our cohort (36%) is overall similar to previous reports that included ACS patients [3,4,19]. The incidence of AKI among patients presenting with AMI has been reported between 8–37% [5,6,7,8,9], yet lower rates, similar to those found in our study, were reported following coronary angiography (not in the setting of AMI) and when preventive measures such as the use of low amounts of contrast agents and routine hydration were implemented [9,12,20]. The increased risk for the development of AKI in this setting among patients with DM is consistent with previous reports [12,13,21,22,23].

The etiologies attributed to the development of AKI in the setting of ACS are diverse and include contrast-induced AKI (CI-AKI) following coronary angiography and especially PCI, hemodynamic instability with compromised renal perfusion (i.e., cardiogenic shock, mechanical complication, or arrhythmias), volume status changes, medications (e.g., nitroglycerine, diuretics, ACE I/ARBs), atheroembolism during PCI, ischemia driven alteration in the function and structure of epithelial cells, non-cardiovascular complications such as infections or inflammatory states as well as patient-associated comorbidity and predisposing risk factors (prior chronic kidney disease, diabetes, increased age, etc.) [24,25,26,27,28,29,30,31]. CI-AKI is a prominent and widely investigated mechanism for AKI in ACS patients following coronary angiography/PCI. In the current study, the rate of PCI was lower among patients that developed AKI, possibly due to differences in presentation and comorbidity (higher rate of chronic kidney disease among patients with AKI). Patients with DM and high glucose levels were reported to be particularly sensitive to these mechanisms [23,32]. DM has been associated with increased reactive oxygen species (ROS) and increased oxidative stress, mediated by increased mitochondrial superoxide and enhanced activity of reduced nicotinamide adenine dinucleotide phosphate (NADPH) oxidase [23]. Furthermore, DM is associated with increased renal oxygen consumption through the enhancement of the load of several ion pumps (e.g., Na+-glucose transporter in renal tubular epithelial cells and Na+-K+-ATPase activity in the medulla) [33]. Additionally, stronger renal vessel constriction and dysfunction of vasoactive substances were reported [34,35,36]. Moreover, various immunological alterations (e.g., infiltration of immune cells into the kidney and release of proinflammatory factors and upregulation of cytokines such as IL-1 (interleukin-1), IL-18, IL-6, IL-33, TGF-β (tumor growth factor-beta), IFN-c (interferon-gamma), and TNF-α (tumor necrosis factor-alpha)) among patients with DM, could directly or indirectly facilitate AKI (particularly CI-AKI) [23]. Different signaling pathways control the pathological changes of renal cells in AKI. Those that were reported to be related to DM are PKB/FoxO, PKB/mTOR/p70S6, p38 MAPK, JNK, and NF-kappa B [23].

Considering the abovementioned mechanisms, the increased incidence of AKI among patients with DM, as found here and elsewhere [12,13,21,22,23], is quite clear. Nevertheless, the similar predictors and particularly the outcomes of AKI in the setting of ACS in patients with vs. without DM, as observed in the current study could be surprising. It seems that the consequences of AKI, when it occurs in the setting of ACS, are so significant that they overshadow other significant risk factors such as DM, which become much less substantial in this setting. This is further supported by AKI being the strongest predictor of mortality with nearly identical HR, both in patients with and without DM, and in the lack of interaction between DM and AKI in the prediction of mortality in the entire cohort adjusted to potential confounders. In addition, AKI could express or be a consequence of other severe complications that cause worse outcomes (e.g., cardiogenic shock, malignant arrhythmia, mechanical complications). Furthermore, it may imply similarity in the mechanisms of AKI between patients with and without DM. Moreover, the worse outcomes of DM patients with ACS could be mediated by the increased risk for AKI hence further emphasizing the great importance of preventive measures for AKI in this group of patients. The independent predictors of AKI in the current study are overall consistent with previously reported risk scores, with the most prominent parameters being CKD, older age and CHF [10,12,28].

The significantly increased mortality rates among patients with AKI versus patients without in the setting of ACS and following PCI are consistent with previous reports [5,6,7,8,9,10,11,25,37,38]. However, the mortality rates among ACS patients who developed AKI in our study are somewhat higher than those reported in most previous studies (up to 44%). This difference probably stems from the non-selective “all-comer” population, including patients with cardiogenic shock and post-resuscitation enrolled in our study. Furthermore, patients were enrolled from ICCUs, and only patients with ACS, while many other studies enrolled patients after PCI (including stable, often ambulatory patients without ACS). Our study also included patients treated two decades ago when outcomes of ACS were worse. Moreover, the definition of AKI used by us was not the most lenient one, as previously mentioned, hence probably excluding the easiest cases with best outcomes.

## 5. Limitations

The current study was retrospective and observational (although patients were enrolled prospectively), and thus shares the limitations of such a design. However, the registry reflects real-life data of all-comers (rather than that of a highly selective population from a clinical trial) which increases the robustness and representativity of our findings. Several unaccounted confounders such as contrast volume (as a cause of contrast-induced AKI), prevention measures, the management of AKI or of predisposing factors (e.g., fluid challenge, mechanical circulatory support), and adherence with treatment recommendations were not reported in the registry. The occurrence of AKI was defined retrospectively by the ACSIS CRF and was not screened prospectively, which might introduce some bias. Furthermore, it was not based on one uniform definition, and rather, a mixture of definitions with some adjustments was used based on existing data and in order to minimize other types of bias which might have introduced some inaccuracies and limited comparability with other studies. Finally, data regarding mortality causes are not reported in the ACSIS registry.

## 6. Conclusions

Patients with DM are at much greater risk for AKI when admitted with ACS. However, the outcomes of AKI, once developed in the setting of ACS, are similar between diabetics and none-diabetics, and the independent predictors of AKI among patients with and without DM are virtually identical. These findings help clinicians in risk stratification as well as making therapeutic decisions specifically regarding measures that affect the risk of AKI according to the diabetes status in order to implement intervention for prevention of AKI (e.g., amount of contrast material, hydration, maintaining adequate perfusion, potentially nephrotoxic medications, etc.).

## Figures and Tables

**Figure 1 jcm-10-04931-f001:**
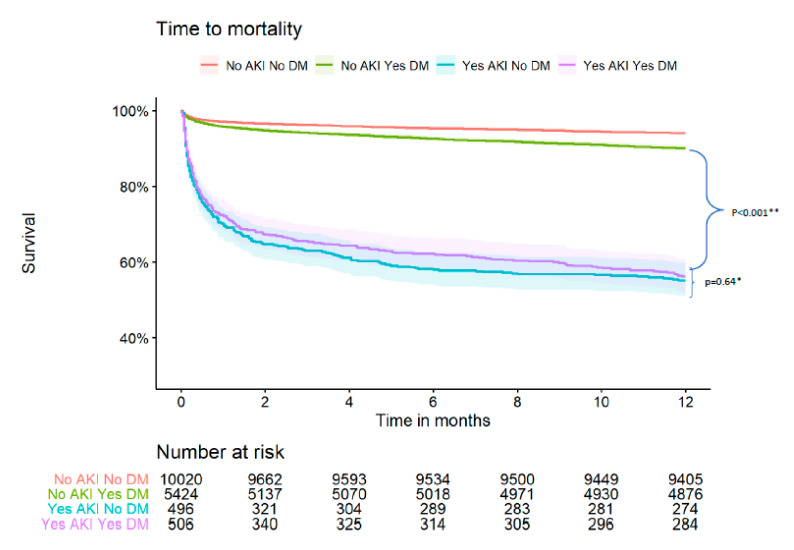
Kaplan Meier survival curve by DM status and AKI. * *p* for comparison between the following groups: AKI_ DM vs. AKI_no-DM (comparison between diabetic patients with AKI vs. non-diabetic patients that developed AKI). ** *p* for comparison between the following groups: AKI_DM vs. no-AKI_DM (comparison between diabetic patients with AKI vs. diabetic patients that did not develop AKI).

**Figure 2 jcm-10-04931-f002:**
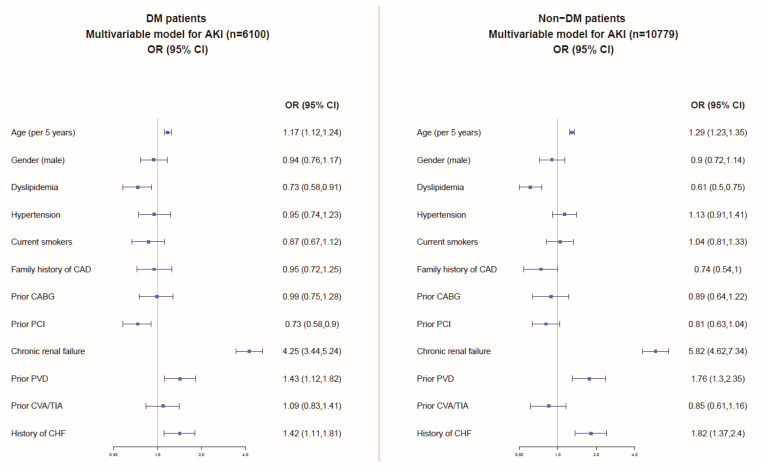
Multivariable model for the prediction of AKI among patients with and without DM. CHF: congestive heart failure; CAD: coronary artery disease; MI: myocardial infarction; TIA: transient ischemic attack; CVA: cerebrovascular attack; PVD: peripheral vascular (arterial) disease; PCI: percutaneous coronary interventional; CABG: coronary artery bypass graft surgery.

**Table 1 jcm-10-04931-t001:** Baseline characteristics by DM status and AKI.

	Overall	no-AKI_no_DM	no-AKI_DM	AKI_ no-DM	AKI_DM	*p* *	*p* **
*n*	16,879	10,276	5587	503	513		
**Baseline characteristics and demographics**							
Age, years (median [IQR)	64 (54, 74)	61 (52, 72)	66 (57, 74)	78 (69, 83)	73 (65, 80)	<0.001	<0.001
Gender (male)	13038 (77.2)	8263 (80.4)	4065 (72.8)	363 (72.2)	347 (67.6)	0.13	0.015
Higher education/ academic	1455 (29.5)	955 (32.9)	438 (24.3)	34 (35.4)	28 (20.3)	0.015	0.34
Marital status: married	6951 (78.4)	4157 (79.2)	2471 (79.1)	135 (61.6)	188 (68.6)	0.13	<0.001
Dyslipidemia	10968 (65.2)	6053 (59.1)	4304 (77.3)	246 (49.1)	365 (71.4)	<0.001	0.003
Hypertension	10035 (59.6)	5087 (49.6)	4190 (75.2)	341 (67.8)	417 (81.4)	<0.001	0.002
Current smokers	6303 (37.5)	4384 (42.8)	1704 (30.7)	117 (23.4)	98 (19.5)	0.15	<0.001
Diabetes mellitus	6100 (36.1)	0 (0.0)	5587 (100.0)	0 (0.0)	513 (100.0)	NA	NA
Family history of CAD	4072 (26.0)	2753 (28.3)	1193 (23.7)	54 (11.6)	72 (16.1)	0.062	<0.001
BMI (kg/m^2^), (median [IQR])	27 (25, 30)	27 (24, 29)	28 (25, 31)	26 (24, 29)	27 (25, 31)	<0.001	0.049
Prior MI	5275 (31.3)	2646 (25.8)	2224 (39.9)	179 (35.7)	226 (44.2)	0.007	0.064
Prior CABG	1655 (9.8)	724 (7.0)	774 (13.9)	61 (12.1)	96 (18.8)	0.005	0.003
Prior PCI	4765 (28.3)	2363 (23.0)	2115 (38.0)	116 (23.1)	171 (33.6)	<0.001	0.057
Chronic renal failure	1832 (10.9)	559 (5.4)	815 (14.6)	207 (41.2)	251 (49.0)	0.014	<0.001
PVD	1393 (8.3)	513 (5.0)	675 (12.1)	83 (16.5)	122 (23.9)	0.004	<0.001
s/p CVA/TIA	1373 (8.1)	596 (5.8)	631 (11.3)	60 (11.9)	86 (16.8)	0.034	<0.001
History of CHF	1363 (8.1)	479 (4.7)	651 (11.7)	104 (20.7)	129 (25.4)	0.088	<0.001
Grace score > 140	1451 (15.0)	576 (9.9)	631 (19.0)	116 (50.2)	128 (44.3)	0.2	<0.001
Earliest creatinine (mg/dL) (median [IQR])	1.00 (0.9, 1.2)	1 (0.85, 1.1)	1 (0.83, 1.3)	1.7 (1.40, 2.3)	1.8 (1.4, 2.6)	0.12	<0.001
**Medical therapy prior to admission**							
Aspirin	7021 (47.8)	3427 (38.4)	3117 (63.5)	196 (47.5)	281 (64.6)	<0.001	0.7
Clopidogrel	1356 (9.4)	591 (6.7)	683 (14.2)	31 (7.5)	51 (11.8)	0.05	0.19
ACE-I	3163 (31.3)	1387 (23.0)	1567 (44.2)	81 (31.8)	128 (43.5)	0.006	0.87
ARB	1216 (12.5)	501 (8.6)	617 (18.1)	23 (9.2)	75 (26.0)	<0.001	0.001
Beta blockers	5335 (37.2)	2625 (30.3)	2322 (48.3)	171 (42.3)	217 (50.0)	0.03	0.52
Statins	6601 (47.4)	3245 (38.7)	2956 (62.5)	149 (38.1)	251 (59.1)	<0.001	0.17
Calcium channel blockers	2989 (21.5)	1380 (16.4)	1338 (28.8)	104 (26.1)	167 (39.4)	<0.001	<0.001
Nitrates	1531 (11.2)	670 (8.0)	685 (15.0)	78 (19.6)	98 (23.4)	0.2	<0.001
Hypoglycemic agents	3488 (23.1)	34 (0.4)	3192 (62.9)	2 (0.5)	260 (58.2)	<0.001	0.054
Diuretics	2066 (17.5)	818 (11.5)	987 (24.5)	101 (28.9)	160 (43.0)	<0.001	<0.001

CHF: congestive heart failure; CAD: coronary artery disease; BMI: body mass index; MI: myocardial infarction; TIA: transient ischemic attack; CVA: cerebrovascular attack; ACE: angiotensin converting enzyme; ARB: angiotensin receptor blockers, PVD: peripheral vascular (arterial) disease; PCI: percutaneous coronary interventional; CABG: coronary artery bypass graft surgery. * *p* for comparison between the following groups: AKI_ DM vs. AKI_no-DM (comparison between diabetic patients with AKI vs. non-diabetic patients that developed AKI). ** *p* for comparison between the following groups: AKI_DM vs. no-AKI_DM (comparison between diabetic patients with AKI vs. diabetic patients that did not develop AKI.).

**Table 2 jcm-10-04931-t002:** In hospital complications, reperfusion therapy and treatment upon discharge by DM status and AKI.

	Overall	no-AKI_no_DM	no-AKI_DM	AKI_ no-DM	AKI_DM	*p* *	*p* **
In-hospital complications							
CHF mild-moderate (Killip-2)	1544 (9.2)	720 (7.0)	570 (10.2)	106 (21.5)	148 (29.1)	0.007	<0.001
Pulmonary oedema (Killip-3)	1099 (6.5)	360 (3.5)	419 (7.5)	155 (30.9)	165 (32.2)	0.7	<0.001
Cardiogenic shock (Killip-4)	569 (3.4)	200 (1.9)	124 (2.2)	137 (27.3)	108 (21.1)	0.03	<0.001
Hemodynamically significant RVI	88 (0.8)	39 (0.6)	22 (0.6)	10 (4.0)	17 (5.6)	0.5	<0.001
Reinfarction	223 (1.3)	106 (1.0)	70 (1.3)	24 (4.8)	23 (4.5)	0.95	<0.001
Post MI angina	793 (4.7)	463 (4.5)	246 (4.4)	33 (6.6)	51 (10.0)	0.07	<0.001
Stent thrombosis (definite/probable/possible)	79 (0.7)	46 (0.7)	28 (0.7)	2 (0.8)	3 (1.0)	1	0.87
Free wall rupture	59 (0.3)	36 (0.4)	15 (0.3)	6 (1.2)	2 (0.4)	0.27	0.95
Tamponade	43 (0.3)	25 (0.2)	12 (0.2)	4 (0.8)	2 (0.4)	0.66	0.76
VSD	20 (0.1)	8 (0.1)	5 (0.1)	2 (0.4)	5 (1.0)	0.47	<0.001
MR moderate—severe	323 (1.9)	125 (1.2)	96 (1.7)	48 (9.6)	54 (10.6)	0.69	<0.001
Pericarditis	107 (0.6)	72 (0.7)	23 (0.4)	4 (0.8)	8 (1.6)	0.4	0.002
Sustained VT (>125 bpm)	247 (1.5)	132 (1.3)	54 (1.0)	30 (6.0)	31 (6.0)	1	<0.001
Primary VF	311 (1.8)	225 (2.2)	55 (1.0)	18 (3.6)	13 (2.5)	0.43	0.003
Secondary VF	126 (0.7)	62 (0.6)	28 (0.5)	20 (4.0)	16 (3.1)	0.56	<0.001
New A. Fib.	895 (5.3)	400 (3.9)	265 (4.7)	123 (24.5)	107 (20.9)	0.2	<0.001
High degree (2–3 degree) AV block	382 (2.3)	197 (1.9)	106 (1.9)	37 (7.4)	42 (8.2)	0.7	<0.001
Asystole	359 (2.1)	138 (1.3)	81 (1.4)	67 (13.3)	73 (14.3)	0.74	<0.001
Stroke	103 (0.6)	42 (0.4)	36 (0.6)	11 (2.2)	14 (2.7)	0.73	<0.001
Bleeding	238 (1.4)	99 (1.0)	72 (1.3)	33 (6.6)	34 (6.6)	1	<0.001
Blood transfusions	170 (3.1)	57 (1.8)	54 (2.6)	23 (18.9)	36 (22.4)	0.57	<0.001
**Coronary angiography and PCI**							
PCI	10,308 (61.1)	6631 (64.5)	3196 (57.2)	237 (47.1)	244 (47.6)	0.94	<0.001
STEMI	7659 (45.4)	5087 (49.5)	2114 (37.9)	243 (48.3)	215 (41.9)	0.1	<0.001
Coronary angiography	11,428 (87.2)	7068 (89.7)	3874 (85.7)	230 (70.6)	256 (68.4)	0.6	<0.001
**Medical therapy upon discharge**							
Aspirin	15,557 (94.4)	9665 (95.5)	5163 (94.4)	357 (80.2)	372 (83.2)	0.28	<0.001
P2Y12	11,926 (72.9)	7504 (74.6)	3923 (72.3)	227 (51.6)	272 (60.7)	0.008	<0.001
Statins	13,946 (85.2)	8612 (85.6)	4777 (88.0)	256 (58.3)	301 (67.3)	0.007	<0.001
ACE-I/ARB	12,029 (74.2)	7139 (71.9)	4412 (82.2)	237 (51.9)	241 (52.9)	0.8	<0.001
Beta blockers	12,873 (79.9)	7910 (80.1)	4416 (82.6)	255 (57.4)	292 (65.0)	0.002	<0.001

CHF: congestive heart failure; CAD: coronary artery disease; BMI: body mass index; MI: myocardial infarction; TIA: transient ischemic attack; CVA: cerebrovascular attack; ACE: angiotensin converting enzyme; ARB: angiotensin receptor blockers, PVD: peripheral vascular (arterial) disease; PCI: percutaneous coronary interventional; CABG: coronary artery bypass graft surgery; VT: ventricular tachycardia; VF: ventricular fibrillation; RVI: right ventricular infarction. * *p* for comparison between the following groups: AKI_ DM vs. AKI_no-DM (comparison between diabetic patients with AKI vs. non-diabetic patients that developed AKI). ** *p* for comparison between the following groups: AKI_DM vs. no-AKI_DM (comparison between diabetic patients with AKI vs. diabetic patients that did not develop AKI.).

**Table 3 jcm-10-04931-t003:** Patient outcomes by DM status and AKI.

	**Overall**	**no-AKI_no-DM**	**no-AKI_DM**	**AKI_ no-DM**	**AKI_DM**	***p* ***	***p* ****
**30-day outcomes**							
Major adverse cardiac events	2376 (14.1)	1221 (11.9)	744 (13.4)	202 (40.3)	209 (40.8)	0.92	<0.001
mortality	805 (4.8)	290 (2.8)	227 (4.1)	148 (29.7)	140 (27.4)	0.45	<0.001
re-hospitalization	2868 (18.9)	1697 (18.1)	1001 (19.9)	83 (23.4)	87 (23.8)	0.99	0.09
reinfarction	286 (1.9)	150 (1.7)	92 (1.8)	21 (5.0)	23 (5.2)	1	<0.001
Angina	322 (4.0)	189 (3.9)	123 (4.3)	5 (3.6)	5 (2.8)	0.91	0.4
CABG	1397 (8.4)	785 (7.8)	556 (10.1)	30 (6.0)	26 (5.2)	0.69	<0.001
**1-year outcomes**							
mortality	1580 (9.6)	597 (6.0)	540 (10.0)	222 (44.8)	221 (43.7)	0.78	<0.001

CABG: coronary artery bypass graft surgery. * *p* for comparison between the following groups: AKI_ DM vs. AKI_no-DM (comparison between diabetic patients with AKI vs. non-diabetic patients that developed AKI). ** *p* for comparison between the following groups: AKI_DM vs. no-AKI_DM (comparison between diabetic patients with AKI vs. diabetic patients that did not develop AKI.).

## Data Availability

The data that support the findings of this study are available from the ACSIS registry organization, but restrictions apply to the availability of these data, which were used under license for the current study, and so are not publicly available. Data are, however, available from the authors upon reasonable request and with permission of the ACSIS registry organization.

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
