# Peer review of "Acute Kidney Injury Following Admission with Acute Coronary Syndrome: The Role of Diabetes Mellitus"

_jcm, 2021, doi:10.3390/jcm10214931_

Round 1
Reviewer 1 Report
The manuscript has been largely improved, both for English language editing and for data presentation. Nevertheless I still can not catch the innovative message it should bring.
Author Response
Reviewer #1
The manuscript has been largely improved, both for English language editing and for data presentation. Nevertheless, I still can not catch the innovative message it should bring.
Response: we thank the reviewer for his review sessions of our manuscript. To further support and emphasize the innovative message and in accordance with the comments of the additional reviewers. We have added
Additional analyses (elaborately specified in the responses to the other reviewers below) were added to our manuscript (divided the study to patients with and without DM and compared between both groups – baseline characteristics [supplementary appendix table 4]), and outcomes: AKI and mortality [supplementary appendix figure 1]). As well as an interaction between AKI and DM in a multivariate model prediction of mortality among patients with ACS (supplementary appendix table 3).
All these further result in a very extensive and detailed evaluation of the association between DM and AKI in the context ACS and further support the finding of the dominance of AKI as a prognostic marker among patients with ACS, irrespective of DM status. This has significant clinical and pathophysiological implications.
Reviewer 2 Report
My doubts have been resolved. Now the article is clearer.
To make the article even more interesting, I would go and calculate the p of interaction between diabetes and aki, in predicting the endpoints.
Author Response
Reviewer #2
My doubts have been resolved. Now the article is clearer.
To make the article even more interesting, I would go and calculate the p of interaction between diabetes and AKI in predicting mortality following ACS
Response: we thank the reviewer for his/her efforts and for acknowledging the improvements introduced to are manuscript. Following this additional comment, we added an additional analysis evaluating for the interaction between DM and AKI in predicting endpoint (supplementary table 3).
Additionally, multivariable analysis adjusted to potential confounders (supplementary appendix table 3), shows no statistically significant interaction between DM and AKI in the prediction of 1-year mortality (p=0.1).
We believe this further supports our findings that outcomes of AKI, once developed in the setting of ACS, are similar irrespective of the DM status.
This was also added to the discussion section: …lack of interaction between DM and AKI in the prediction of mortality in the entire cohort, adjusted to potential confounders.
Reviewer 3 Report
The authors report a case control study of the prevalence of acute kidney injury in acute coronary syndrome, by dividing it into subset of diabetes mellitus versus no diabetes mellitus. They report similar associations of covariates associated with AKI in patients with DM versus no DM. Furthermore, they report presence of AKI as a strong predictor of mortality irrespective of DM status. Here are my major comments. 1. The authors attempt to exhibit the data in multiple different ways, which is confusing. Recommend either reporting the study as a cohort, demonstrating the impact of DM on AKI, or divide the study into DM versus no DM and focus on AKI and mortality as an outcome. 2. The discussion section has a heavy emphasis on contrast induced AKI. The study does not report contrast amount and volume, hence recommend shifting the discussion to the findings of the study.
Author Response
Reviewer #3
The authors report a case control study of the prevalence of acute kidney injury in acute coronary syndrome, by dividing it into subset of diabetes mellitus versus no diabetes mellitus. They report similar associations of covariates associated with AKI in patients with DM versus no DM. Furthermore, they report presence of AKI as a strong predictor of mortality irrespective of DM status.
Response: we thank the reviewer for this summary of our study and main findings.
Here are my major comments. 1. The authors attempt to exhibit the data in multiple different ways, which is confusing. Recommend either reporting the study as a cohort, demonstrating the impact of DM on AKI, or divide the study into DM versus no DM and focus on AKI and mortality as an outcome.
Response: following this comment by the reviewer we added a comparison between patients with vs. without DM (supplementary appendix table 4). In addition, we focused on the outcomes requested by the reviewer: AKI (table 4) and 1-year mortality (supplementary appendix figure 1.)
Results section
Additionally, a comparison of the baseline characteristics between patients with and without DM is presented in the supplementary appendix 4.
Furthermore, a multivariable analysis with 1-year mortality as the outcome (supplementary appendix figure 1) showed similar predictors among patients with and without DM, specifically AKI was the strongest independent predictor in both groups (with DM: OR=3.55, 95%CI: 2.99-4.21, p<0.001, without DM: HR=3.69, 95%CI: 3.09-4.4, p<0.001) with similar HR in both.
we also added an additional analysis evaluating for the interaction between DM and AKI in predicting endpoint (supplementary table 3).
Additionally, multivariable analysis adjusted to potential confounders (supplementary appendix table 3), shows no statistically significant interaction between DM and AKI in the prediction of 1-year mortality (p=0.1).
- The discussion section has a heavy emphasis on contrast induced AKI. The study does not report contrast amount and volume, hence recommend shifting the discussion to the findings of the study.
Response: we thank the reviewer for this comment. Indeed, the contrast volume was missing in our study as acknowledged in the limitations section and now further elaborated and stressed:
“Several unaccounted confounders such as contrast volume (as a cause of contrast induced AKI)”
CI AKI is an important cause of AKI in the setting of ACS accounting for a significant proportion of cases and reviewed in our literature review, hence we relate to it in the discussion section. However, following the reviewer comment we shortened the discussion sections relating to it (specifically we omitted the extended discussion of the mechanisms of CI-AKI). And elaborated on the findings more specifically associated with our findings, specifically also focusing on the abovementioned additional findings asked by the reviewer
This is further supported by AKI being the strongest predictor of mortality with nearly identical HR both in patients with and without DM and in the lack of interaction between DM and AKI in the prediction of mortality in the entire cohort adjusted to potential confounders.
Round 2
Reviewer 3 Report
The authors have satisfactorally addressed the concerns.
Author Response
we thank the reviewer for her/his insights and comprehensive review of our manuscript.
we are pleased for that the reviewer found our responses to be satisfactory.
we have performed additional minor revision of the manuscript as requested.